# Multiwalled Carbon Nanotubes-Modified Metallic Electrode Prepared Using Chemical Vapor Deposition as Sequential Injection Analysis Detector for Determination of Ascorbic Acid

**DOI:** 10.3390/nano13071264

**Published:** 2023-04-03

**Authors:** Abdalghaffar M. Osman, Abdulmajeed Hendi, Nadir M. A. Osman

**Affiliations:** 1Chemistry Department, King Fahd University of Petroleum and Minerals (KFUPM), Dhahran 31261, Saudi Arabia; 2Interdisciplinary Research Center (IRC) for Advanced Materials, King Fahd University of Petroleum and Minerals (KFUPM), Dhahran 31261, Saudi Arabia; 3Physics Department, Fahd University of Petroleum and Minerals (KFUPM), Dhahran 31261, Saudi Arabia; 4Interdisciplinary Research Center (IRC) for Hydrogen and Energy Storage, King Fahd University of Petroleum and Minerals (KFUPM), Dhahran 31261, Saudi Arabia

**Keywords:** CNTs-Ag electrode, chemical vapor deposition, strong CNTs adhesion to metal surface, direct electrical contact, differential electrolytic potentiometry

## Abstract

A carbon nanotubes modified silver electrode (CNTs-Ag) was prepared via catalytic chemical vapor deposition and characterized. The morphology, crystallinity, elemental composition, and other quality parameters of the prepared electrode were investigated using scanning electron microscopy (SEM), transmission electron microscopy (TEM), X-ray diffraction (XRD), X-ray photoelectron spectroscopy (XPS), and Raman techniques. The characterization results revealed the modification of the silver metal surface with CNTs of good characteristics. A sequential injection analysis (SIA) system was developed for studying the reaction of ascorbic acid with KIO_3_ using the prepared CNTs-Ag electrode. Electrodes were polarized with both direct current (DC) and periodic square wave (SW). Various experimental conditions affecting the differential electrolytic potentiometric (DEP) peak such as current density, SW bias value, and flow rate were appraised. Under the optimum conditions, good linear responses for ascorbic acid were obtained in the range of 60.0–850.0 µM for both types of polarization with detection limits of 14.0–19.0 µM. The results obtained showed that the periodic polarization method was more sensitive than DC polarization and the electrode response was faster. Ascorbic acid in pharmaceutical tablets was determined with satisfactory results using this method. The prepared CNTs-based electrode exhibited good performance for a long period of use. The method is simple, rapid, and inexpensive for routine analysis.

## 1. Introduction

Electrochemical techniques have advantages over other techniques such as spectrometry and chromatography in food, biological, pharmaceutical, and clinical applications. These include high precision and sensitivity, selectivity, cost effectiveness, defined reaction time, environmentally friendly, miniaturization, and minimum sample pretreatment [1,2]. Nanomaterials have been extensively used to develop advanced electrochemical sensors and biosensors for a wide range of applications [3,4,5]. Due to a combination of exceptional properties including electrical configuration, chemical structure, large surface-to-mass ratio, and adsorption capacity, carbon nanotubes (CNTs) have been widely used as electrode material for electrochemical sensing and biosensing [6,7]. The unique electronic structure of CNTs makes them an excellent choice to mediate heterogeneous transfer of electrons in electrochemical reactions [8]. In addition, their high surface area and good capability for adsorption makes CNTs suitable electrode material for stripping techniques [9,10]. Moreover, CNTs provide an excellent surface to immobilize various moieties and they are easily functionalized to suit different applications [11]. As electrode material, CNTs are commonly attached to the surface of conventional bare electrodes using either drop/spin-casting of CNTs well-dispersed solution, or spraying techniques [12]. Modifications of bare electrodes using these methods result mostly in agglomeration and formation of nanotubes clusters [13]. To avoid that, CNTs are usually mixed with nafion [14], oxidized, esterified, modified with polysaccharides and biomolecules [15], or converted into polymer composites [16,17,18]. Bare electrodes modified with CNTs through the afore-mentioned methods are subject to the detachment of the CNTs, which limits the electrode efficiency. In addition to that, CNTs are not uniformly aligned, but distributed randomly within the electrode matrix, limiting their maximal potential to transfer electrons [19].

To build CNTs-based electrodes with the desired properties, an effective approach to modify metallic electrodes with CNTs is to synthesize the CNTs directly on the surface of a metallic substrate. The significance of this approach is to overcome the shortcomings of the randomly distributed CNTs on the bare electrode surface as well as providing good contact in between [20]. Among the commonly used techniques for producing CNTs, catalytic chemical vapor deposition (CCVD) is the preferred route for quite pure CNTs as compared to the other two techniques: arc discharge and laser ablation methods [21,22]. CCVD is a simple and cost-effective approach for producing CNTs with a more controlled architecture at ambient pressure and relatively lower temperatures [23,24]. In CVD method, CNTs grow on catalyst nanoparticles that are homogeneously distributed over a substrate by using a suitable carbon source at high temperature and an inert environment. Compared to other methods utilized for modifying bare electrodes with CNTs, in situ CVD fabrication of such electrodes produces electrodes with aligned CNTs that have direct electrical contact with the substrate that improve the electron transfer [25]. CNTs’ growth on metallic substrate rather than attaching them to the bare electrode surface using the classical methods mentioned earlier offers a stronger adhesion in the metal CNTs film interface and therefore minimizes the contact resistance and facilitates the electron transfer, which is critical in electrochemical applications [26]. Thus far, few in situ CVD fabrications of CNTs-modified electrodes have been reported in the literature.

Because of its simplicity, versatility, and flexibility, sequential injection analysis (SIA) is widely utilized in the field of flow chemistry for quantitative analyses [27]. SIA has been used to successfully determine a variety of analytes in food and pharmaceutical samples [28,29,30] utilizing electrochemical detection methods [31]. Electrochemical detection has gained importance in SIA systems because it provides a fast response with a sensitivity that is independent of the flow-through cell compared to some other detection methods [32]. Potentiometric detectors are simple to build in the lab at a minimal cost, and they are easy to incorporate into SIA manifolds [32], providing rapid and precise potentiometric measurements [33].

Ascorbic acid (known as vitamin C) is an antioxidant which acts against free radical-induced diseases. It is one of the most common electroactive biological species that are easily oxidized, which constitutes the basis of its electrochemical determination [34]. Therefore, electrochemical methods are the most common for ascorbic acid detection and quantitation in different applications, especially with the development of using nanomaterials to enhance sensitivity and selectivity [34,35,36].

Differential electrolytic potentiometry (DEP) has been used as an indicating system for the quantitation of ascorbic acid in pharmaceutical and food samples using conventional titrations [37,38], and in SIA systems [39,40]. SW polarized working electrodes exhibited more sensitivity and lower detection limit due to the great acceleration of the electrode response and significant decrease in the electrode deactivation [41].

The purpose of this work is to bridge the gap in the production of robust electrodes modified with high-quality aligned CNTs with good adhesion to the bare substrate to be utilized for long periods. The DEP technique with different types of polarization was employed to increase the sensitivity and maintain the activity of the electrode.

## 2. Experimental

### 2.1. Chemicals, Reagents, and Preparations

Ascorbic acid (99.5%), potassium iodate (99.5%), and sulfuric acid from Sigma Aldrich were used without purification. A stock aqueous solution of ascorbic acid (0.10 mol/L) was prepared prior to any analysis and used within 8 h to prepare the other required diluted concentrations. A 0.10 mol/L stock solution of potassium iodate was prepared and used for the preparation of other diluted solutions. A total of 0.05 mol/L of sulfuric acid was also prepared as supporting electrolyte. Three tablets of vitamin C (Redoxon) were ground to powder. A 250.0 mL solution including 76.0 mg of the powder was prepared using deionized water and later used for the analysis of ascorbic acid. The same pharmaceutical sample was analyzed using normal titration against standardized 0.0466 mol/L iodine solution using starch indicator, according to the United States pharmacopeia (USP) [42].

### 2.2. Instrumentation

The surface morphology was studied using a Tescan–VELA3 scanning electron microscope. Transmission electron microscope (JOEL-2100F) was used to prove the formation of multi-walled nanotubes and observe more details. The crystallinity of CNTs was examined using an X-ray diffractometer (Rigaku Ultima IV) equipped with Cu Kα X-ray source (λ = 0.15406 nm). The scan rate and acquisition 2 theta range were 0.5 degree/min and 20 to 90, respectively. The chemical analysis was performed using X-ray photoelectron spectrometer (Thermo Scientific, ESCALAB 250 Xi, Waltham, MA, USA) equipped with an Al Kα (1486.6 eV) X-ray source. Thermo Avantage software was employed for XPS curve fitting. The pressure inside the analysis chamber was 7 × 10^−11^ mbar). The C 1s peak centered at 285 eV was used as a reference for calibrating binding energies of the obtained XPS spectra. Raman spectrum was acquired from LabRam, HORIBA Scientific Raman spectrometer using 633 wavelength excitation laser with energy of 2.54 eV with gratings (1800 lines/mm) and eV and an acquisition range from 100 to 3000 cm^−1^.

### 2.3. Preparation and Characterization of CNTs-Coated Silver Electrode

The coating of silver wire with CNTs was achieved in a CVD reactor using the pyrolysis of acetylene and then the CNTs growth over iron nanoparticles as a growth catalyst according to the procedure mentioned elsewhere [43]. The prepared CNTs-Ag electrode was characterized using different analytical techniques to investigate the morphology of the electrode surface and other quality parameters of the grown CNTs such as crystallinity, graphitization, and atomic composition.

### 2.4. SIA Configuration and Procedure

The developed SIA system is illustrated in Figure 1. The system consists of FIAlab-3500 unit (FIAlab Instruments, Seattle, WA, USA) with Alitea peristaltic pump and microsyringe pump and additional separately programmable microsyringe pump (J-KEM Scientific, St. Louis, MO, USA). The plexiglass flow cell was designed to accommodate two indicator CNT-Ag electrodes. The analog signals conversion into digital ones is achieved with a LabJack interface and the whole system is operated and controlled using FIAlab 5.0 software. Reservoirs containing solutions of sulfuric acid and ascorbic acid are linked to the first and second syringe pumps, respectively, whereas KIO_3_ solution is propelled using the peristaltic pump. The manifold was initially flushed with sulfuric acid (0.05 mol/L) for a period of time. To obtain the optimal SIA conditions, the value of the current density (µA/cm^2^), the % bias of the SW, and the flow rate (µL/s) of KIO_3_ solution were optimized. Initially, sulfuric acid and 0.02 mol/L KIO_3_ solution were introduced at constant flow rates of 150.0 and 100.0 µL/s, respectively, while ascorbic acid solution (0.28 mmol/L) was pumped at a flow rate of 80 µL/s. The applied current density and percent bias varied from 10–63 µA/cm^2^ and 0–50%, respectively, and the potential (∆E) in triplicate was measured each time. After optimizing both current density and percent bias, the flow rate of KIO_3_ was varied from 50.0–180.0 µL/s. At optimum values of all parameters studied, the potential difference (∆E) in triplicate was measured for a series of ascorbic acid standard solutions in the range 0.06–0.85 mmol/L.

## 3. Results and Discussion

### 3.1. Morphological and Structural Characterizations of the CNTs Electrode

The SEM images of the morphology of the bare Ag and CNTs-Ag electrodes are represented in Figure 1A,B. Figure 1B exhibits bundles of aligned nanotubes, while the TEM image (Figure 1C) revealed the formation of multi-walled CNTs with few walls displaying an inner diameter of 10 nm and outer one of approximately 15 nm. The XRD diffractogram (Figure 2) of the CNTs-coated electrode revealed two characteristic sharp diffraction peaks at around 25° and 42° that corresponded to the C (002) and C (100) crystallographic planes of CNTs [44]. The Ag exhibited the main crystallographic planes Ag (111), Ag (200), Ag (220), Ag (311), and Ag (222) at 38°, 44°, 64°, 77°, and 83°, respectively [45].

The XPS surface analysis was performed to further elucidate the chemical composition of the CNTs-Ag electrode. The survey spectrum for the electrode surface presented in Figure 3A showed the presence of the characteristic signal of C. No signal is observed for Ag, implying the total coverage of Ag substrate by a dense layer of the grown CNTs. For the C 1s spectrum, it can be observed in Figure 3B that the deconvolution of the spectrum resulted in a single strong peak centered at BE of 284. 4 eV, which is attributed to the sp^2^ carbon [46]. Interestingly, the presence of such a single peak indicates the formation of well-graphitized sheets for the prepared CNTs and supports the findings of the XRD and Raman. Similarly, the fitting of the O 1s spectrum (Figure 3C) also showed a single peak positioned at 531.90 eV that might be ascribed to the OH of the adsorbed water on the electrode surface [46,47].

The tangential G band, disorder-induced D band and the second-order harmonic G′ band are the most significant peaks that provide information about the quality of CNTs [47]. Raman spectrum of the as-synthesized CNTs is shown in Figure 4. The strong G-band at 1575 cm^−1^ corresponds to the in-plane vibration of the graphite lattice and indicates the formation of CNTs with good graphitization properties [48]. The G′-peak at around 2640 cm^−1^ confirms the high degree purity of the prepared CNTs since its strength is more sensitive to SP^2^ carbon and dramatically enhanced by CNTs [49]. The reason behind that is G′-peak arises from a two-phonon process, as presence of more impurities does not allow for the coupling effect necessary for the two-phonon process [50,51]. Although many reasons might account for the enhancement of the D peak at 1326 cm^−1^ such as the presence of some amorphous carbon or any sort of distortion in the SP^2^ configuration of the synthesized CNTs, it is most likely that the main reason behind the enhancement of the peak intensity is the high density aligned CNTs, which was proved using the SEM image and supported by the interpretation of other Raman peaks and further confirmed through XRD and XPS analyses [52].

### 3.2. Performance of CNTs-Ag as SIA-DEP Indicator Electrodes

Ascorbic acid is one the most common electroactive biological species that are easily oxidized, which constitutes the basis of its electrochemical determination. In this method, ascorbic acid is oxidized to dehydroascorbic acid with iodate in a sulfuric acid medium [53]. In DEP, the signal results from the potential difference between the anodically and cathodically polarized electrodes. At the beginning of the analysis, IO_3_^−^ ions are reduced at the cathode, and a stable base potential for the system is established. When ascorbic acid is introduced into the flowing stream, it reacts with IO_3_^−^ while oxidation product and I^−^ ions are formed. Consequently, the potentials of the two polarized electrodes change, generating a measurable signal.

### 3.3. Optimization of DEP Parameters

The fabricated CNTs-Ag electrode was applied as a DEP indicating system for the determination of ascorbic acid with both DC and SW (biased and bias-free) periodic polarization. Different parameters such as current density, percent bias of SW, and flow rate were optimized. The current density applied to the indicator electrodes was in the range of 10–63 µA/cm^2^. It was noticed that the obtained peaks were sharp with low intensity at low current density; however, as the current density increased, the peaks became more intense and started to broaden and distort (Figure 5A,B). No significant increase in the peak intensity was noticed beyond the current density value of 40 µA/cm^2^ as shown in Figure 5C. Then, a current density of 21 µA/cm^2^ was selected as an optimum value and applied for all other measurements. In electrode periodic polarization, Figure 6A,B explain the increase of the potential peak intensities by applying more percent bias of up to 40%. However, applying a bias beyond 40% had no significant effect on the peak intensities, Figure 6C, which could be attributed to the deleterious effect of the DC bias component that develops at the higher SW bias [54]. Moreover, it was also noticed that high percent biased caused peak distortion and broadening. It was clear from Figure 6C that the bias-free (0%) of the SW resulted in the lowest potential intensity of the peak.

In addition, applying of high biased SW caused peak broadening, which may result in less accuracy when determining the reaction endpoints. Thus, a maximum of 10% biased SW was chosen as an optimum bias considering both the intensity and sharpness of the peak. Comparing the performance of DC and square-wave polarization, it is found that the latter mode of polarization offers better sensitivity and accuracy and faster electrode response due to the reason that periodic potentials stabilize within few seconds and remain [41].

### 3.4. Characteristics of CNTs-Ag Electrode

In addition to the advantages of polarizing the indicator electrodes, the well-control of the CVD parameters in this study led to the production of CNTs-Ag electrodes with aligned CNTs bundles and well-built graphitic sheets. This resulted in CNTs of high metallic conductivity and hence a high ability to enhance the heterogeneous electron transfer kinetics [8], which, in turn, results in fast establishment of the potential equilibrium. Moreover, the CVD produced CNTs film that is adherent to the silver metal and scratch-proof, with a direct and strong electrical contact that enhances the electrode performance and durability. A probable explanation for the CNTs’ high adherence to the Ag metal surface is the interdiffusion adhesion between CNTs and FeNPs catalyst film on one side, and the interfacial adhesion due to the interatomic interactions between the FeNPs and Ag metal on the other side [55]. These interactions could be electrostatic, chemical, or Van der Waals types. All the above-mentioned characteristics represent the advantages of the CVD fabrication of CNTs-modified electrodes over those modified with randomly distributed CNTs.

### 3.5. Optimization of SIA Parameters

The flow rate in the system is one of the most important contributing factors to consider in SIA since it affects the analyte dispersion, reaction with the reagent, and signal strength. The KIO_3_ flow rate was varied between 50 and 175 µL/s for both DC and SW polarization modes in the SIA system. The maximum potential was obtained at KIO_3_ flow rate of around 80 µL/s, as presented in Figure 7. At flow rate less than this value, the drop in the potential might be ascribed to the decrease of the analyte dispersion and as a result, a lower amount reacted. Beyond this point, increasing the flow rate results in reducing the contact time of the sample components and they thus pass into the detection cell before they fully react.

### 3.6. Analytical Performance for Ascorbic Acid Determination

At the optimum conditions, the peaks obtained for the reaction between ascorbic acid and KIO_3_ using the SIA system were sharp and highly reproducible in both DC and SW polarization as shown in Figure 8A and Figure 9A, respectively. However, higher signals, as seen in Figure 9A, have resulted in the case of SW polarization. The higher sensitivity of the SW mode is attributed to the continuous reversal of the square-wave signal that prevents the buildup of analyte films and/or the adsorption of impurities on the indicator electrode surface and makes it fully active throughout the measurement period [54]. Calibration curves for ascorbic acid solutions spanning the range 60.0–850.0 µM for both DC and SW polarization modes were found to follow a linear relation with a square of correlation coefficient (R^2^) in more than 0.995 in all cases, as represented in Figure 8B and Figure 9B, respectively. The results also showed good reproducibility expressed in terms of percent relative standard deviation (%RSD). Other analytical parameters are provided in Table 1. According to the results obtained in the table, there is an agreement in the values of ascorbic acid determined using the methods described in this work with the standard method in the USP with values of %RSD in the range of 2.4–12.6. The DEP technique using CNTs-Ag electrodes either with DC or periodic polarization for the determination of ascorbic acid is more sensitive, reaching the equilibrium in a short time, and has lower limit of detection than the normal potentiometry when there is no current is passed through the indicator electrode [43]. Ascorbic acid in pharmaceutical tablets of vitamin C (Redoxon) was determined under the optimum conditions of SIA/SW-DEP and the resulting potential peaks are shown in Figure 10. The obtained result agreed with normal titration against standardized 0.0466 mol/L iodine solution using the starch indicator, according to the USP the standard method for ascorbic acid determination [42] as shown in Table 1. The CNTs-Ag electrode prepared via CVD proved to be robust and produced almost the same potential value (204.0 ± 12.3) over a period of 40 days, as shown in Figure 11. This is mainly due to the advantage of using in situ growth of CNTs at high temperature using CVD, which results in strong adhesion of the CNTs layer to the Ag metal. In addition, the periodic polarization used in the DEP technique offers a clean electrode surface during and after the reaction by removing any material adsorbed to it. A simple DEP titration of ascorbic acid was performed using Ag electrodes and bare CNTs-Ag at 20% SW biased to show the impact of the CNTs in enhancing peak signal. It was found that the signal generated from the CNTs-Ag electrode was much stronger, sharp, and corresponded to the expected volume of the titrant, compared to that generated from the bare Ag electrode as shown in Figure 12.

The performance of the CNTs-Ag electrode was compared to that of other electrodes that include CNTs in their composition and used for the electrochemical determination of ascorbic acid. As presented in Table 2, although most of the electrodes used have lower detection limits (LOD) compared to the prepared CNTs-Ag electrode, they are composed of three or four kinds of materials including the CNTs without an advantage of selectivity. Moreover, there is no proof that a single electrode can work for a long period of time with almost the same performance, which gives the CNTs-Ag electrode the privilege of being used for routine work. In addition, the CNTs layer might be functionalized to selectively determine different kinds of analytes.

## 4. Conclusions

In summary, preparation of CNTs-Ag electrode through CVD was described. Characterization of the fabricated electrode confirmed the growth of aligned, well-graphitized, and crystalline MWCNTs. CNTs-Ag electrodes exhibited excellent performance as an SIA-DEP indicating system for the determination of ascorbic acid. SW periodic polarization of the indicator electrodes showed certain advantages over the DC mode. The developed SIA-DEP system was reliable and the potential peaks obtained at the optimum conditions were reproducible. The CNTs-Ag electrode showed a fast response, stable measurements, and attained equilibrium in a short time because of the characteristics of the grown CNTs and strong adherence to the silver metal surface that offers good electrical contact. The CNTs-Ag electrode displayed a stable performance for a long time of use. The developed method for the determination of ascorbic acid combined the advantages of the prepared electrode as an indicating system, periodic polarization, and SIA, which finally produced a reliable analytical method that requires a reduced use of reagents.

## Data Availability

Not applicable.

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
