# Peer review of "Multiwalled Carbon Nanotubes-Modified Metallic Electrode Prepared Using Chemical Vapor Deposition as Sequential Injection Analysis Detector for Determination of Ascorbic Acid"

_nanomaterials, 2023, doi:10.3390/nano13071264_

Round 1
Reviewer 1 Report
The manuscript entitled “Multiwalled Carbon Nanotubes-Modified Metallic Electrode Prepared by Chemical Vapor Deposition as Sequential Injection Analysis Detector for Determination of Ascorbic Acid” is an original manuscript presenting the CNT-enhanced properties of sensors. The positive effects of the carbon nanotubes for electrochemical systems is quite well know, but this works brought new findings for this particular application as demonstrated on the example of the probing the organic acids.
The text is well written but some issues must be resolved before further processing:
1) First of all, the XPS C1s curve fitting was done using wrong parameters. The fitting of the XPS C1s signal is not appropriate. The MWCNTs are conductive and their sp2 carbon electrons signal has strong asymmetric shape. The authors used common Gaussian-Lorentzian line shape and thus the had to add many additional functions including C-O, etc. Indeed, in very old publications, the researchers already reported the C1s curve fitting using that wrong model employing Gaussian-Lorentzian line shape for all peaks. Indeed, nowadays all appropriate models for Graphene, CNTs, graphite, etc. took into account the fact that the C1s signal coming from SP2 carbon of conductive phase has strongly asymmetric line shape. Please see the following references describing the XPS C1s curve fitting for these cases use similar method referencing to that works:
https://doi.org/10.1016/j.apsusc.2022.153681
https://doi.org/10.1016/j.tca.2018.03.017
2) What is atomic composition determined by XPS? Please report C and O at.% concentrations
3) What was the O1s shape and position?
4) Can You incorporate the EDX results?
Reviewer 2 Report
The paper is devoted for determination of ascorbic acid using multiwalled carbon nanotubes modified silver electrodes. The topic is generally interesting, however the paper contain unexplained places (below) and need major revisions.
All abbreviations should be explained by first using, for example LOD, RSD Table 2, USP at line 113.
The abstract and conclusions should be rewritten in more informative and logic way.
The aim of the paper should be more clearly indicated.
Fig. 1A should be commented in the paper text.
Line 162 ‘’with good coverage of the Ag substrate’’ it is really observed in Fig. 1B?
Figs. 8-11 and Table should be more discussed.
It would be useful if you try your system for detection of other materials, not only ascorbic acid.
More comparisons with results presented in literature should be added to the table 2.
The text is unclear at line 203.
All typos should be corrected, for example line 231: percent bias. of up to, However
English need minor corrections.
Reviewer 3 Report
In this study, Ag electrode was modified with multiwalled CNTs through chemical vapor deposition. The prepared electrode was characterized by various microscopy and spectroscopy techniques. The performance of the electrode for the quantitative analysis of ascorbic acid was also tested in a sequential injection analysis apparatus.
The work is interesting and in general it is well-written. There is a large portion of experimental work. The results are clearly presented and adequately discussed.
Besides editing (see some examples below), I think that the authors should perform some revisions by taking account the following:
1) As it is discussed by authors, besides CNTs, graphitized sheets and most likely amorphous carbon were also detected on the surface of the metallic electrode. Thus perhaps the title could be revised and instead of CNT it would be better to use the term carbonaceous nanoparticles or low purity CNT.
2) In the abstract it is mentioned that “The carbon nanotubes (CNTs) film was very adherent to the silver metal and scratch-proof.” I don’t think that any of these statements (and especially the scratch-proof) can be supported by the presented results. If this is a conclusion from a previous work the authors can mention it in the results or in the introduction with the respective reference. In any case this sentence should be removed from the abstract.
3) It would be interesting for the reader to add some discussion about the performance of the “control/reference” electrode, that is, the pure Ag electrode. Is there any evidence (from previous work or other literature) that the CNTs increase the performance of the Ag electrode?
Typos and other comments
Line 89 “of” is missing
Line 184 “of” is missing
Table 2. Use capital letter in the first word of Table 2 caption.
Round 2
Reviewer 1 Report
First of all, the XPS C1s curve fitting was done using wrong parameters.
Dear authirs, you did not follow my recommendation at all. Your XPS fitting is absolutely wrong. PLEASE READ MY COMMENT AGAIN Simply please see the survey soectrum, you have so low O1s signal, probably 1 at%, while c-O and OCOO contributions together have more than 10at%. This is impossible. Please correct lineshapes, read the references provided and resubmit.Otherwise you always need to add the C-O contributions thatvare absent in your material.
The fitting of the XPS C1s signal is not appropriate. The MWCNTs are conductive and their sp2 carbon electrons signal has strong asymmetric shape. The authors used common Gaussian-Lorentzian line shape and thus the had to add many additional functions including C-O, etc. Indeed, in very old publications, the researchers already reported the C1s curve fitting using that wrong model employing Gaussian-Lorentzian line shape for all peaks. Indeed, nowadays all appropriate models for Graphene, CNTs, graphite, etc. took into account the fact that the C1s signal coming from SP2 carbon of conductive phase has strongly asymmetric line shape. Please see the following references describing the XPS C1s curve fitting for these cases use similar method referencing to that works:
https://doi.org/10.1016/j.apsusc.2022.153681
https://doi.org/10.1016/j.tca.2018.03.017
Author Response
See the attached file, please

Reviewer 2 Report
Authors make proper corrections and I suggest to publish the paper as it is.
Author Response
We thank the reviewer for his/her positive comments
Reviewer 3 Report
The authors responded adequately to most of my comments. I suggest acceptance of the manuscript
Author Response

(The authors gave the same response as above.)

Round 3
Reviewer 1 Report
The C1s curve fitting was improved. The manuscript was revised accordingly and the paper can be accepted in present form.